# Correspondence Heuristic and Filter-Empowerment Heuristic: Investigating the Reversed Gender Achievement Gap in a Sample of Secondary School Students in Saudi Arabia within the Framework of Educational and Learning Capital

**Heidrun Stoeger** [1,*], **Norah Almulhim** [2] and **Albert Ziegler** [3]

1. Department of Educational Sciences, University of Regensburg, D-93053 Regensburg, Germany
2. Department of Education, King Faisal University, Al Hofuf 31982, Saudi Arabia
3. Department of Psychology, Friedrich-Alexander-Universität Erlangen-Nürnberg, D-90478 Nürnberg, Germany
* Correspondence: heidrun.stoeger@ur.de

**Abstract:** The reversed gender achievement gap in the Kingdom of Saudi Arabia (KSA) in favor of girls developed in a societal environment that still predominantly favors men. The finding illustrates how equity gaps at higher systemic levels may not translate seamlessly to lower systemic levels. We propose that this lack of correspondence between systemic levels regarding equity gaps can be explained by the fact that more exogenous learning resources (educational capital) selectively reach girls' actiotopes, enabling them to build more effective endogenous learning resources (learning capital) and, in particular, effective action repertoires. To investigate this assumption, we introduce a filter-empowerment heuristic and contrast it with a simpler correspondence heuristic. We employ the filter-empowerment heuristic and the education and learning capital approach to investigate the reversed gender achievement gap in a sample of secondary school students in the KSA. We test four hypotheses: (a) Girls have higher academic achievement than boys. (b) Girls have more educational capital and (c) more learning capital than boys. (d) Educational and learning capital mediate the relationship between gender and academic achievement. The hypotheses were tested with a sample of 2541 ninth-grade students from 55 KSA schools. The four hypotheses were confirmed and support a filter-empowerment heuristic rather than a correspondence heuristic.

**Keywords:** equity gap; gender; reversed gender achievement gap; learning and educational capital; Saudi Arabia; secondary school

## 1. Introduction

In education, equity gaps are defined as differences between groups that violate notions of equity [1]. Equity gaps thus describe "the inequitable treatment of diverse groups" [2] (p. 377). Such inequities have been found in virtually all social fields in education [3,4] such as social class, race, ethnic groups, gender, or sexual orientation. According to Leithwood [2], "concerns about inequity have never been more widespread or garnered more public energy than they do now" (p. 377). In this paper, we focus on equity gaps with respect to gender [5–7].

A wide range of gender equity gaps has been found, including gender gaps in confidence [8], in political media coverage [9], in leader emergence [10], and in various types of performances [11,12]. Gender gaps in education not only occur in many forms, but they have been found at all systemic levels. Using Bronfenbrenner's [13,14] terminology, systemic levels include the chronosystem (i.e., environmental changes that occur over the lifespan; see [15,16]), macrosystems (i.e., attitudes and ideologies of the culture; see [17,18]), exosystems (i.e., the extended family and neighborhood, see [19,20]), mesosystems, (i.e.,

interactions of an individual's micro systems; see [21,22]), and microsystems (i.e., an individual's parents, siblings, peers, and teachers; see [23,24]). Many authors assume that gender gaps at these systemic levels may cause gender gaps at the individual level in interaction with psycho-biological factors [20,25,26]. Such reasoning is found, for example, when gender gaps at higher systemic levels are held responsible as causes of emergence of gender gaps at lower systemic levels. A well-known example is the assumption that gender gaps on the individual level can be attributed to patriarchal social structures that favor men [27,28]. However, the reverse line of reasoning, that gender gaps at lower systemic levels cause gender gaps at higher systemic levels is equally possible [29]. A well-known example is the assumption that biologically determined risk-taking by men leads to gender gaps at higher systemic levels [30].

## 2. Reversed Equity Gaps and the Advancement of the Correspondence Heuristic into a Filter-Empowerment Heuristic

### 2.1. The Correspondence Heuristic

The correspondence heuristic assumes that equity gaps on one systemic level translate into equity gaps in the identical direction on other systemic levels. A corollary of the correspondence heuristic is that if there is no gender gap on one system level, that system level cannot be responsible for a gender gap on another system level. We would like to illustrate the correspondence heuristic with two prototypical cases.

For example, there are large gender gaps in some STEM fields, while there are much smaller gender gaps in others [31,32]. Suppose a researcher uses the correspondence heuristic to analyze whether the mesosystem of work environments underlies gender gaps in STEM fields. If this were the case, then according to the correspondence heuristic the magnitude of gender differences in STEM fields should also be reflected in corresponding differences in work environment characteristics in these fields. Following the correspondence heuristic, Su and Rounds [32], for example, were able to show that varying degrees of gender gaps in STEM fields corresponded to varying degrees of person-orientation and subject-orientation of the work environment in these STEM fields.

An example for the corollary of the correspondence heuristic, that if there is no gender gap on one system level, that system level cannot be responsible for a gender gap on another system level, would be the following: According to the Gender Inequality Index (GII), gender equality has practically been achieved in Switzerland [33], while females in Switzerland are still significantly underrepresented in research and innovation in STEM. According to the correspondence heuristic, the three (currently measured) indicators recorded in the GII (reproductive health, empowerment, and labor market) are presumably not responsible for the (current) gender gap in research and innovation in STEM. Instead, it would be assumed that the indicators recorded in the GII will have an effect on research and innovation in STEM with a time lag. This would mean that the currently lower contribution of women in research and innovation in STEM is a consequence of the past state of the GII indicators. If this assumption were correct, then, based on the current GII, it would be expected according to the correspondence heuristic that the gender gap with regard to research and innovation in STEM will soon be closed. Current evidence does not, however, suggest this is the case.

Overall, the correspondence heuristic has proven to be quite fruitful. However, in our view it also has its limits and requires further development and specification. In particular, the phenomenon of reversed equity gaps is difficult to reconcile with it. The term reversed gender gap was chosen in light of the predominant expectation "[ . . . ] of what it means to promote gender equality, namely helping girls catch up with boys [34] (p. 3508)".

### 2.2. The Reversed Gender Gap

Equity gaps in education can be highly dynamic, as evidenced, for example, by multiple historical improvements for women [35–38]. Though most equity gaps are still biased against women [39], the situation especially in education has changed substantially [40].

Some authors such as Van Bavel et al. [41] even speak of a completely changed situation: "The gender gap in education has reversed in recent decades in most Western and many non-Western countries" (p. 341). However, such a general assessment is certainly overstated. Consider three of numerous counterexamples. (a) As Van Bavel et al. [41] indirectly admit, gender equality is far from being achieved in many developing countries [42–44]. (b) Kahn and Ginther [45] point to multiple disadvantages for women in the math-intensive science fields, which is particularly evident in their self-selection out of the more lucrative fields in STEM [46,47]. (c) One central goal of education is also empowering individuals for their paths after leaving public educational institutions. Research suggests that women are less well prepared by their education systems for work life [39,48], e.g., when it comes to negotiating salaries [49,50].

At present, it is unclear what causes reversed gender gaps. Although they are now becoming a global phenomenon with regard to educational attainment, "[their] drivers, however, are not well understood and remain largely untested empirically", [51] (p. 2). Moreover, there are only partial explanations [52,53] based on a plethora of individual and environmental variables associated with gender gaps. For example, they include economic growth and socio-economic gaps [54,55]; culture [56]; social norms [57]; competitive environment [58]; school context [59]; differences in the distributions of scholastic performance [51]; family structure [60]; mothers' employment [61]; or fathers' absence [34]. The exemplary list of causes supports the assumption that reversed gender achievement and educational gaps are a complex, multi-causal phenomenon operating at various systemic levels.

Reversed equity gaps are difficult to capture with a simple correspondence heuristic, and the situation is further complicated when differential effects are considered at the same systemic level. For example, consider school performance and confidence measured at the individual level. Using a correspondence heuristic, it is not easy to explain why girls simultaneously rate high on grades and low on confidence [62]. We therefore consider the correspondence heuristic to be in need of refinement and, to this end, present an advancement of this heuristic towards the filter-empowerment heuristic. Introducing the filter-empowerment heuristic, we will first focus on gender gaps that are measured at the individual level. As a case in point, we will focus on school performance.

*2.3. The Filter-Empowerment Heuristic*

The filter-empowerment heuristic aims to explain gender gaps measured at a specific systemic level. An example is a reversed gender achievement gap, which can be operationalized, for example, as better academic performance of girls. The unit of analysis is thus the girls' personal system and its interaction with other systems. The filter-empowerment heuristic states that the resources necessary for successful learning, as reflected in academic achievement, are present and utilized to varying degrees in girls' personal systems.

Two mechanisms are postulated to cause the varying degrees of resources responsible for the emergence of gender gaps: filters and empowerment. A filter (denoting barriers, obstacles, exclusions, etc.) can impede a personal system's ability to access certain resources. Under certain conditions, the systems interacting with a personal system can also actively transport more resources into the personal system and thus empower it. For example, research has repeatedly found that STEM fields are associated with characteristics such as brilliance, self-centeredness, and analytic strength [63,64], which, in turn are more strongly connected to the stereotypical male than the stereotypical female [17]. In this way, the stereotype acts as a filter in STEM against women by curtailing their access to relevant resources and supporting the empowerment of men in STEM. In the terminology of systems theory, the stereotype acts as a repeller for women and as an attractor for men.

## 3. Correspondence Heuristic and Filter-Empowerment Heuristic Applied to the Reversed Gender Achievement Gap in the Kingdom of Saudi Arabia

As mentioned earlier, although the correspondence heuristic has proved quite fruitful in many areas, it is limited in explaining the phenomenon of the reversed gender gap.

In our view, the filter-empowerment heuristic appears more promising. The accuracy of this assumption can only be tested in a setting where there is a clear reversed gender gap. The situation in Arab countries is relevant in this context, especially when considering the Human Development Index (HDI). In 2019, the Human Development Index (HDI) globally for women was significantly lower at an average of 0.71 compared to 0.76 for men [65]. The situation in Arab countries is interesting. Their HDI is 0.85, which classifies them as nations with "very high human development" according to the United Nations [65]. At the same time, the difference in HDI between men and women is 0.10, twice the global average. In fact, the situation in the Kingdom of Saudi Arabia (KSA) is steadily improving in terms of gender equality. However, although the KSA's global gender gap index has improved, from 0.56 in 2007 to 0.64 in 2021, representing an annual index growth rate of 0.87% [66], the KSA is only ranked 127th out of 146 countries in the World Economic Forum's Global Gender Gap Report 2022 [39].

Importantly, however, the gender gap in favor of males does not appear when looking at academic performance. In fact, there is a reverse gender gap, meaning that female students show significantly better academic performance than male students. For example, in the 2018 PISA survey, girls in the KSA scored 54 points higher than boys in reading (compared to a 30-point advantage of girls in the OECD average). In mathematics, girls in the KSA had a 13-point performance advantage over boys (while the OECD average was 492 points for boys, slightly better than girls' 487 points). Similarly, in science, where boys and girls achieve almost identical OECD averages of 490 and 488 respectively, the KSA had a reverse gender gap of 29 points in favor of girls. In fact, the reversed gender gap in the Middle East can be seen among cohorts born in the late 1970s [51].

A reversed gender gap in school performance in a country where there is still a general gender gap in favor of men raises problems for the correspondence heuristic. Using the filter-empowerment heuristic seems more promising. In this context, the first question to be answered is which resources needed for school-relevant learning are more accessible to and better used by girls in KSA.

## 4. Educational and Learning Capital

Ziegler's and Stoeger's [67] educational and learning capital approach [67–70] can be used to investigate these resources. According to the educational and learning capital approach, resources are means to an end. The resources that constitute educational and learning capital consist of all endogenous and exogenous resources that can be used to enable learning processes to succeed and thus increase the likelihood of successful learning processes. The authors postulate five endogenous learning resources (located in the individual) and five exogenous learning resources (located in the environment), respectively, which exhaustively categorize the resources necessary for learning processes according to an analysis by Vialle [71]. They refer to endogenous learning resources alternatively as learning capital. These include organismic learning capital, telic (motivational) learning capital, actional learning capital, episodic learning capital, and attentional learning capital (for definitions refer to Table 1). The authors refer to exogenous learning resources alternatively as educational capital. These include economic educational capital, social educational capital, cultural educational capital, infrastructural educational capital, and didactic educational capital (for definitions refer to Table 1).

**Table 1.** Definitions of the Ten Types of Educational and Learning Capital From Ziegler and Baker [68].

| Type of Exogenous Resource | Definition | Type of Endogenous Resource | Definition |
|---|---|---|---|
| Economic educational capital | Economic educational capital is every kind of wealth, possession, money, or valuable that can be invested in the initiation and maintenance of educational and learning processes. (p. 27) | Organismic learning capital | Organismic learning capital consists of the physiological and constitutional resources of a person. (p. 29) |
| Cultural educational capital | Cultural educational capital includes value systems, thinking patterns, models, and the like that can facilitate—or hinder—the attainment of learning and educational goals. (p. 27) | Telic learning capital | Telic learning capital comprises the totality of a person's anticipated goal states that offer possibilities for satisfying their needs. (p. 30) |
| Social educational capital | Social educational capital includes all persons and social institutions that can directly or indirectly contribute to the success of learning and educational processes. (p. 28) | Actional learning capital | Actional learning capital means the action repertoire of a person—the totality of actions a person is capable of performing. (p. 30) |
| Infrastructural educational capital | Infrastructural educational capital relates to materially implemented possibilities for action that permit learning and education to take place. (p. 28) | Episodic learning capital | Episodic learning capital concerns the simultaneous goal- and situation-relevant action patterns that are accessible to a person. (p. 31) |
| Didactic educational capital | Didactic educational capital means the assembled know-how involved in the design and improvement of educational and learning processes. (p. 29) | Attentional learning capital | Attentional learning capital denotes the quantitative and qualitative attentional resources that a person can apply to learning. (p. 31) |

The unit of analysis of the education and learning capital approach is the actiotope of a student, that is, the personal system of a student and their enacted environment [72]. Figure 1 illustrates the model's assumed causal process, with a constant influx of educational capital into the actiotopes of students. Students can transform educational capital within their actiotopes into learning capital based on their already existing learning capital. Thus, in building effective action repertoires to master school learning requirements, students are, on the one hand, dependent on what educational capital an environment offers. On the other hand, however, students can also actively seek out potential learning environments rich in educational capital on the basis of their learning capital and use them more or less effectively based on their own learning capital. This process, shown schematically in Figure 1, has already been used successfully both to analyze achievement trends in Arab countries [73] and specifically in the KSA [74] as well as to analyze gender differences [75,76].

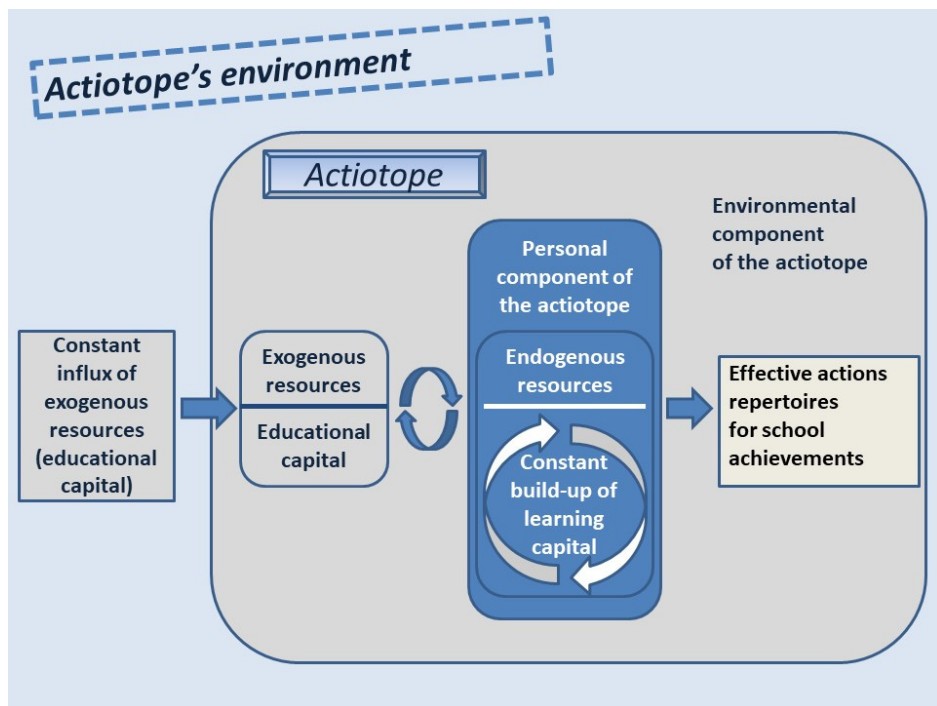

**Figure 1.** Influx of Educational and Learning Capital into Actiotopes and Its Processing into Effective Action Repertoires for School Performance.

## 5. Current Study

We chose the KSA as the setting for our study. Its prevailing patriarchal social structures favoring men [27,28] and the reversed gender achievement gap that has been demonstrated for the KSA in many studies [51] make it particularly suitable for examining the filter-empowerment heuristic (in contrast to the correspondence heuristic). Before going into more detail about our study, we will briefly describe the education system in the KSA.

In the KSA, children of both sexes attend nine years of compulsory education. Children between the ages of three and five can attend kindergarten. However, kindergarten attendance is not a prerequisite for admission to primary education. Primary education lasts six years; intermediate and secondary education last three years each [77]. During their second year of secondary education, students choose a humanities or a natural sciences track. Schooling in the KSA is free, including textbooks [78]. The Ministry of Education issues and manages a unified curriculum and textbooks for all schools [79]. The total number of schools in the KSA is 22,986, of which 80.9% are public schools [80]. After completing secondary school, students can attend technical/vocational colleges or universities [77]. Universities typically require students to pass benchmark tests, such as a general aptitude test (similar to the SAT and ACT in the United States) and standardized academic achievement tests. Post-secondary education is also provided free of charge to Saudi citizens [77]. As in the society as a whole, there is gender segregation. Educational institutions are, with few exceptions, either for males or for females [79].

In our study, we investigated the reversed gender achievement gap in the KSA. Considering earlier research [51,81], we assumed a reversed gender achievement gap in secondary school in our KSA sample. We specified Hypothesis 1 as follows: Girls show better scholastic achievements than boys.

As the correspondence heuristic seems unsuitable to explain a reversed gender gap in school performance in a country where there is still a general gender gap in favor of men, as is the case in the KSA [65], we applied the filter-empowerment heuristic. More specifically, we investigated the differential availability and use of resources (i.e., educational and learning capital) of girls and boys in the KSA, which we assumed would exist on the

basis of the filter-empowerment heuristic and previous research [73,74]. Specifically, we investigated whether the differential possession and use of educational and learning capital can explain the reversed gender achievement gap in the KSA. To this end, we tested three additional hypotheses. Hypothesis 2 was that girls possess more educational capital on average than boys in the KSA. Hypothesis 3 was that girls possess more learning capital on average than boys in the KSA. Hypothesis 4 was that educational and learning capital mediate the relationship between gender and academic achievement in the KSA.

## 6. Method

### 6.1. Participants

In order to test our hypotheses, we asked secondary students in the KSA to fill out a questionnaire. Participants were 2541 ninth-grade students from 55 schools in Al-Ahsa, the largest governorate in the KSA's Eastern Province. Students were on average 14.18 years old (range: 13–15 years, SD = 0.29); 61% were female; and 68% were from schools in urban areas. Twelve percent of students reported that their parents' highest educational attainment was less than high school, 27% reported high school, 42% reported diploma or bachelor's degree, 9% reported master's degree, 4% reported a doctoral-level degree, and 5% of students did not provide valid information about their parents' highest educational attainment.

### 6.2. Procedure

Before data collection, ethics approval was obtained, and a sampling procedure was chosen. First, the approval for the data collection procedure and the instruments of the study was sought from the Ethics Committee of King Faisal University. Then, the Ministry of Education was contacted in order to gain access to the schools and to acquire demographic information. Out of the 112 schools in Al-Ahsa, 55 schools were selected. To do so, a mix of stratified random sampling and convenience non-random sampling was utilized. The first step in sampling was to divide the schools into two subgroups based on the students' gender (in the KSA schools are separated by gender). Next, these two subgroups were divided further into four subgroups, one based on the school location (urban or rural) and the other based on the type of school (private or public). Then, a random sample was taken from each stratum (boys' private school, boys' public school, boys' urban school, boys' rural school, girls' private school, girls' public school, girls' urban school, and girls' rural school). The data were collected from February to April 2019. It was not possible for the authors to personally supervise the entire data-collection process. Teachers were enlisted to supervise students while they completed the questionnaire.

### 6.3. Measures

#### 6.3.1. Educational Capital

Students reported their educational capital by responding to the 20 educational capital items from the Arabic version [82] of the Questionnaire for Educational and Learning Capital (QELC; [83]). The items are divided into five 4-item subscales, measuring the five types of educational capital: economic educational capital, cultural educational capital, social educational capital, infrastructural educational capital, and didactic educational capital. Students answered these items on a six-point Likert-type scale (1 = not at all true; 2 = not true; 3 = rather not true; 4 = rather true; 5 = true; 6 = absolutely true). A sample item for economic educational capital is "My family has enough money to support the development of my academic skills." Cronbach's α of the scale was 0.85.

#### 6.3.2. Learning Capital

Students reported their learning capital by filling out the 20 learning capital items from the Arabic version [82] of the QELC [83] The items are divided into five 4-item subscales that measure the five types of learning capital: organismic learning capital, actional learning capital, telic learning capital, episodic learning capital, and attentional learning capital. These items were answered on the same scale as those for educational capital. A sample

item for organismic learning capital is "I am so physically fit that I can learn and study for school for long periods of time without getting tired." Cronbach's $\alpha$ of the scale was 0.90.

### 6.3.3. Academic Achievement

Students' academic achievement was assessed with their grade-point average (GPA). GPA in the KSA ranges from 0 to 100, with 100 indicating the highest performance. GPAs were provided by students in response to the question "What is your GPA from your last half-year report?" Students' GPA is based on their grades in mathematics, Arabic language, science, English language, social studies, religion, computer science, and family education.

### 6.3.4. Gender

Students' gender was reported by the students. Male was coded as 0, while female was coded as 1.

### 6.4. Plan of Analysis

To test hypotheses 1–3, independent-sample *t* tests were calculated. To test hypothesis 4, two approaches were used to carry out a mediation analysis: The causal-steps approach (to calculate the direct effect of the multiple mediation [84]) and the bootstrapping approach (to calculate the total and specific indirect effect of the multiple mediation [85]).

The causal steps approach [84] was used to estimate the paths in the proposed model. Therefore, multiple linear regression was utilized to test the mediation in four steps (see Figure 2). Step one was to confirm that female students exhibit greater academic achievement than male students (path c). The second step was to confirm that female students have both more educational capital (path $a_1$) and more learning capital (path $a_2$) than male students. The third step was to confirm that both educational capital (path $b_1$) and learning capital (path $b_2$) are still positively related to academic achievement when controlling for the other type of capital and the effect of gender on academic achievement. The fourth step was to confirm that the effect of gender on academic achievement is significantly reduced when controlling for educational capital and learning capital (path c', the effect after controlling for educational capital and learning capital), which indicates mediation. Although this causal-step approach is the most used procedure for testing mediation [85], it does not provide an indirect effect of the multiple mediation and the associated confidence intervals. Hence, the bootstrapping procedure recommended by Preacher and Hayes [85] was also implemented. For this purpose, the process macro for SPSS [86] was used to test the two parts of the multiple mediation model (see [85]): (a) the total indirect effect (i.e., whether both educational capital and learning capital convey the effect of gender on academic achievement), and (b) the specific indirect effect (i.e., whether each of the two capital types conveys the effect). Because it is possible to obtain a significant total indirect effect and specific indirect effects that are not important, and vice versa, we analyzed both types of effects (total and specific) as well as their confidence intervals via bias-corrected confidence-intervals bootstrapping (based on 5000 bootstrap samples).

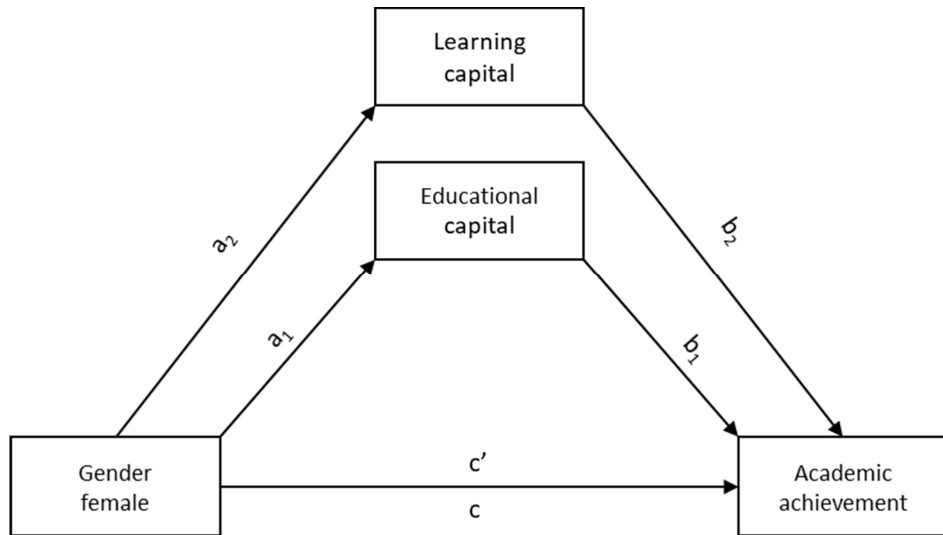

**Figure 2.** Proposed Multiple Mediation Model for the Relationship Between Gender and Academic Achievement, Mediated by Educational Capital and Learning Capital. Path c represents the total effect of gender on academic achievement; path c′ represents the direct effect of gender on academic achievement.

## 7. Results

Before conducting the primary analyses, we assessed the psychometric properties for the study variables (see Table 2). We assessed whether the assumption of normal distribution was met for the variables under investigation based on their skewness. All values for skewness were within an acceptable range (see [87]). Moreover, the correlations between the variables were calculated (see Table 3).

**Table 2.** Psychometric Properties of all Study Variables.

| Variables | M | SD | Range | Skewness |
|---|---|---|---|---|
| Academic achievement | 89.20 | 9.86 | 49–100 | −0.99 |
| Educational capital | 4.61 | 0.78 | 1.40–6.00 | −0.67 |
| Learning capital | 4.68 | 0.80 | 1.05–6.00 | −0.72 |

**Table 3.** Correlation Matrix for all Study Variables.

| Variable | 1 | 2 | 3 |
|---|---|---|---|
| 1. Academic achievement | - | | |
| 2. Female gender | 0.11 | - | |
| 3. Educational capital | 0.30 | 0.09 | - |
| 4. Learning capital | 0.34 | 0.13 | 0.75 |

Note. All correlation coefficients are significant at $p < 0.001$.

To test hypotheses 1–3, the values of the variables were compared between boys and girls. This revealed that girls reported significantly higher levels of academic achievement, educational capital, and learning capital (see Table 4 for means, standard deviations, *t*-test results, and effect sizes).

**Table 4.** T Tests for Comparison Between Girls and Boys Regarding Academic Achievement and Educational and Learning Capital.

| Variable | Girls | | Boys | | *t* | *p* | Cohen's d |
|---|---|---|---|---|---|---|---|
| | **M** | **SD** | **M** | **SD** | | | |
| Academic achievement | 90.07 | 9.37 | 87.83 | 10.44 | 5.49 | <0.001 | 0.226 |
| Educational capital | 4.67 | 0.75 | 4.53 | 0.82 | 4.29 | <0.001 | 0.178 |
| Learning capital | 4.76 | 0.75 | 4.54 | 0.85 | 6.58 | <0.001 | 0.274 |

Next, the proposed multiple mediation model was tested. The standardized regression coefficients for the relationship between gender and academic achievement as mediated by educational capital and learning capital can be found in Figure 3. As expected, female students exhibited higher academic achievement than male students ($\beta = 0.11$, $p < 0.001$). Further, female students exhibited both higher amounts of educational capital ($\beta = 0.09$, $p < 0.001$) and higher amounts of learning capital ($\beta = 0.13$, $p < 0.001$) than male students. Additionally, both greater amounts of educational capital ($\beta = 0.12$, $p < 0.001$) and learning capital ($\beta = 0.24$, $p < 0.001$) predicted greater academic achievement. Calculating the unstandardized specific indirect effects of educational capital (B = 0.20, $p < 0.01$, 95% CI [0.09, 0.37]), learning capital (B = 0.65, $p < 0.001$, 95% CI [0.42, 0.93]), and the total indirect effect of both educational and learning capital (B = 0.86, $p < 0.001$, 95% CI [0.56, 1.17]) revealed that a significant proportion of the effect of gender on academic achievement was mediated by educational capital and learning capital (as zero did not occur between their interval ranges), thus confirming the hypothesis that educational capital and learning capital mediate the relationship between gender and academic achievement.

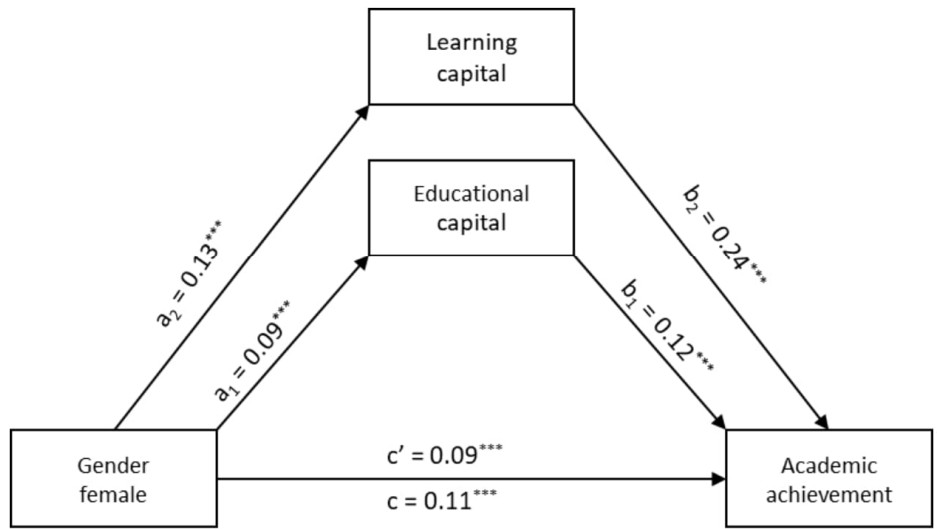

**Figure 3.** Standardized Regression Coefficient for the Relationship Between Gender and Academic Achievement, Mediated by Educational Capital and Learning Capital. *** $p < 0.001$.

## 8. Discussion

The phenomenon of discrimination against women runs through history [35–38] and has, however, been reversed in many fields [40,41]. This is especially true for academic achievement. While equity gaps in education generally refer to violations of educational equity, reversed gender equity gaps in education specifically refer to violations of educational equity in which the historically dominating direction of disadvantage for women has been reversed [40,41]. Of course, each individual equity gap and each individual reversed gender equity gap has its own history and unique causal constellations. However, it is not implausible that there may be commonalities across individual equity gaps in education [1].

This is also the basis of what we dubbed the correspondence heuristic, the belief that equity gaps at a systemic level can be explained by corresponding equity gaps at other systemic levels. As we have argued, however, reversed equity gaps are difficult to capture with a simple correspondence heuristic. The limits of the correspondence heuristic motivated us to develop this heuristic further into the filter-empowerment heuristic. Its core is the assumption that differential availability and use of relevant resources cause equity gaps between groups. Similar to the correspondence heuristic, the filter-empowerment heuristic appears theoretically plausible and is supported by empirical research. For example, studies (e.g., [70,88]) show that different resources are partly responsible for gender inequality in science and professional life. Specifically, it has been shown that women are on average less able to devote time, energy, and commitment to their careers (e.g., due to more extensive childcare and housework), which can lead to lower productivity.

In the current study, we chose the gender achievement gap in the KSA as an interesting test case for the filter-empowerment heuristic. Although the country has made progress in developing gender equity in recent years, at the macro-structural level, women are still disadvantaged [39]. Interestingly, however, the gender gap in favor of males does not appear when looking at academic performance. In fact, there is a reverse gender gap, meaning that female students show significantly better academic performance than male students [51]. The filter-empowerment heuristic opens promising perspectives for tackling the problem of similarities and differences between individual equity gaps [1]. This problem can now be concretized into a comparison of those resources available and used in the actiotopes of individuals that enable successful learning processes. To establish comparability, the education and learning capital approach of Ziegler and Stoeger [67–69] was used. Within the framework of the actiotope model, this approach can distinguish between the exogenous resources that enter a learners' personal systems and the endogenous resources that are necessary for successful learning at school. However, we would like to point out here that the filter-empowerment heuristic can also be combined with other theoretical approaches that establish comparability across equity gaps. The choice of the learning and educational capital approach was motivated by five considerations: (a) it specifies both endogenous and exogenous resources; (b) it is comprehensive and exhaustive [71]; it has already been successfully applied (c) in the target context of our study, the KSA, as well as in other cultures, and it has been used (d) to explain gender differences; finally, (e) a measurement instrument based on the learning and educational capital approach validated in an Arab country was already available.

In the current study, we were able to replicate the findings about a reversed gender achievement gap in the KSA [51,81] in our sample (Hypothesis 1). Considering the assumptions of the filter-empowerment heuristic and the education and learning capital approach, we expected that girls' academic achievement advantages in the KSA would be attributable to girls there having access to more exogenous and endogenous resources for successful learning than their male peers. In other words, in comparison to boys in the KSA, we expected to see a greater inflow of educational capital into KSA girls' actiotopes and a concomitantly greater resulting generation of additional learning capital within the girls' actiotopes. In line with this expectation, three specific hypotheses were tested. In Hypothesis 2, we assumed that more educational capital would flow into girls' actiotopes than into boys' actiotopes in Saudi Arabia. This hypothesis was confirmed. In Hypothesis 3, we assumed that girls were able to build better learning capital than boys due to the richer educational capital being provisioned into their actiotopes. This hypothesis was also confirmed. This finding leads to an interesting further speculation about a possible virtuous circle or Matthew effect that we were not able to test in our cross-sectional study. Since girls have more learning capital, they should in turn become even better able to use the educational capital that enters their actiotopes. Furthermore, teachers or parents, for example, could also be expected to respond to girls' better learning potentials due to their increased learning capital and accordingly provide them with more educational capital. Moreover, as girls' learning capital develop, they should become more effective

at also seeking out more educational capital themselves. If these assumptions are correct, gender gaps should widen over time. Such cumulative effects, which lead to a widening of gender achievement gaps over time, have already been observed, for example, in the study by Berendes and colleagues [89], in which the achievement advantage of girls in reading compared to boys increased over the secondary school years. Taken one step further, such cumulative effects of educational capital that favor the accumulation of learning capital, which in turn leads to a better influx and utilization of further educational capital, could also offer a possible explanation for the even more general Matthew effect [90,91].

In addition to the two hypotheses about differences in the availability of educational and learning capital between girls and boys, we hypothesized that educational and learning capital are responsible for the reversed gender achievement gap. Specifically, Hypothesis 4 asserted that education and learning capital mediate the relationship between gender and academic achievement. This hypothesis was also confirmed.

The three confirmed hypotheses (Hypotheses 2–4) form a good starting point for further systematic investigations of the filter-empowerment heuristic. In fact, a complete test of the filter-empowerment heuristic would address at least three more unclarified problems, which we will refer to as the profile problem, the filter-empowerment problem, and the correspondence problem. The correspondence problem can also be understood as a specification of the correspondence heuristic we addressed above. We briefly introduce each problem as a point of departure for future research.

Profile problem: In our study, we have not differentiated which types of educational and learning capital and which combinations of these types of capital cause the reverse gender gap, we have only shown that the combined capitals have explanatory power. In a next step, the different types of learning and educational capital should be differentiated, and it should be investigated whether individual types of learning and/or educational capital have a particularly strong influence or which profiles of types of educational and learning capital are responsible for the reverse gender gap. Studies that examine explanatory variables for gender gaps that can be classified as educational capital and learning capital suggest that specific causal influences can be attributed to different types of learning and educational capital. For example, in a study in the United Arab Emirates, a neighboring country that is culturally similar to the KSA, Ashour [92] was able to show a very broad network of underlying factors for the reverse gender gap. In the study, both family and school influences, which can be considered exogenous resources (i.e., educational capital), and various non-cognitive traits, which can be classified as endogenous resources (i.e., learning capital), have been shown to be influencing factors. The education and learning capital approach allows for even more detailed and holistic analyses of which categories of exogenous and endogenous variables are responsible for the emergence of the reversed gender gap. A more detailed knowledge of educational and learning capital profiles is particularly important for better understanding the emergence of equity gaps and, in a next step, optimally managing resources and supporting disadvantaged individuals and groups.

Filter-empowerment problem: In our empirical study, we analyzed the situation after exogenous resources had already entered the students' actiotopes. However, equity gaps do not only arise in the actiotope, that is, at the interface of individuals and their environments. Rather, they arise at various systemic levels that include family [60,61], school [59], peer group [23,24], cultural system [56], and chronosystem [15,16]. In future research, the influx of resources from different systems needs to be traced to get a better understanding about the factors creating equity gaps.

Correspondence problem: The correspondence problem is more specific than the correspondence heuristic. It postulates that it is not enough to investigate correlations of variables on different systems levels (like the correspondence heuristic does), but that it is essential to trace back the differential influx of resources and resource profiles into an actiotope. The contradictory results of the following two studies show the necessity for such an approach: Ghasemi and Burley [93] found that in some countries with a smaller adult

gender gap students exhibited higher gender differences in mathematics-relevant affect [93]. This finding does not seem to fit the observation of Else-Quest et al. [94] who reported that gender equity in school enrollment, women's share of research jobs, and women's parliamentary representation were the best predictors of reversed gender achievement gaps in math. For a better understanding of these contradictory results an important research strategy would be to not only investigate correlations of variables on the macro-structural level and the individual level (the actiotope), but to systematically investigate how macro-structural differences precisely cause a differential influx of resources into actiotopes. Such differential resource flows into actiotopes, controlled by filters and reinforcements could reveal how asynchronies, i.e., opposing equity gaps, are possible at different systemic levels and, in particular, how special superiorities can develop despite overall disadvantages.

## 9. Limitations and Future Research

Although our study contributes to explaining the reversed gender achievement gap in the KSA, it has several limitations. First, we did not test the full range of the filter-empowerment heuristic in our empirical study. Our study only shows that the combined types of educational and learning capital have explanatory power; it does not take a differentiated look at which types of educational and learning capital and which combinations of these cause the reverse gender gap. It also only starts after the influx of resources into students' actiotopes already had occurred and does not investigate how differences on specific systemic levels precisely cause differential influxes of resources into the actiotopes. Future research should attend to the three problems mentioned above, the profile problem, the filter-empowerment problem, and the correspondence problem.

Second, our sample cannot claim representativeness. Although, we cannot identify any obvious bias, our sample is nonetheless a mixture of a random and a convenience sample. Furthermore, it is not clear to what extent the results of our study and the application of the filter-empowerment heuristic to the reversed gender achievement gap are transferable to other equity gaps, educational settings, and cultures. Future studies should try to replicate our findings in relation to other equity gaps and in other educational settings and cultures.

Third, although the filter-empowerment heuristic makes broad claims about the dynamics of the gender achievement gap, our initial study only examined the parts that are testable in a cross-sectional design. Future studies should use longitudinal designs and interventions to investigate how differential resources flowing into actiotopes at different systemic levels influence the development of equity gaps and reversed equity gaps.

## 10. Conclusions

A better understanding of the mechanisms responsible for equity gaps is essential for creating more equity in education and more broadly in societies. We introduced the filter-empowerment heuristic as an analytical framework to comprehensively examine mechanisms behind equity gaps. In our study, we employed the filter-empowerment heuristic and the education and learning capital approach to investigate the reversed gender achievement gap in the KSA. We showed that the influx of educational and learning capital is essential for the reversed gender achievement gap in our sample. However, the results of our study are only a first step towards a better understanding of the underlying mechanisms. Findings from future studies can contribute to an even more comprehensive understanding and thus, in the long run, make possible optimizations at different system levels and facilitate the creation of more equity.

**Author Contributions:** Conceptualization, H.S. and A.Z.; methodology, H.S., N.A. and A.Z.; software, H.S., N.A. and A.Z.; validation, H.S., N.A. and A.Z.; formal analysis, N.A.; investigation, N.A.; resources, H.S. and N.A.; data curation, H.S. and N.A.; writing—original draft preparation, H.S. and A.Z.; writing—review and editing, H.S., N.A. and A.Z.; visualization, N.A.; supervision, H.S.; project administration, N.A.; funding acquisition, not applicable. All authors have read and agreed to the published version of the manuscript.

**Funding:** This research received no external funding.

**Institutional Review Board Statement:** The study was conducted in accordance with the Declaration of Helsinki, and approved by the Ethics Committee of Deanship of Scientific Research, King Faisal University (approval Code: KFU-REC/2018-12-1, Date: G12/05/2018).

**Informed Consent Statement:** Informed consent was obtained from all subjects involved in the study.

**Data Availability Statement:** Data can be requested from the authors.

**Conflicts of Interest:** The authors declare no conflict of interest.

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
