# Peer review of "Correspondence Heuristic and Filter-Empowerment Heuristic: Investigating the Reversed Gender Achievement Gap in a Sample of Secondary School Students in Saudi Arabia within the Framework of Educational and Learning Capital"

_education, doi:10.3390/educsci12110811_

Round 1
Reviewer 1 Report
Such an interesting frame to consider gender equity issues! I admit it was new learning for me, and I appreciate that. As a newcomer to this theory, I did find the article less accessible than it perhaps could be, in other words, I found a high use of jargon. Not sure if that can be reduced however. I will just speculate that your audience may be reduced because of it.
Author Response
Dear Reviewer 1:
Thank you very much for your very positive feedback. We are pleased that you like our manuscript and that you have rated it positively in all areas in the reviewer form.
Regarding your comment:
"Such an interesting frame to consider gender equity issues! I admit it was new learning for me, and I appreciate that. As a newcomer to this theory, I did find the article less accessible than it perhaps could be, in other words, I found a high use of jargon. Not sure if that can be reduced however. I will just speculate that your audience may be reduced because of it."
As you mentioned, it is difficult to reduce technical terms and make the manuscript more accessible without losing important nuances. Nevertheless, we have gone through the whole manuscript again intensively and tried to make it more accessible and use fewer technical terms. We hope that this will help to improve the manuscript even further.
Thank you again for taking the time to review our manuscript!
Reviewer 2 Report
This could be a very interesting contribution to analyze the gender gap in education. However, the paper needs much work to be a quality improvement of data.
1) From the title, it is not possible to know if the paper deals with higher education, secondary education, primary education... Should be clarified.
2) From the title, the reader would expect to find the analysis of the evolution (heuristic), but the experience is based on collection of data in 2019.
3) From the title, reversed gender achievement gap, one would expect to see a more positive achievement. From the text, it looks like there are optimistic data, but the gap is not explained as closed.
4) STEM includes very different areas. Literature shows that in general there is a higher success for male students in Maths and Physics, but in Biology the numbers are the contrary. If this is not the case in Saudi Arabia, a better explanation is needed.
5) The article includes assumptions from 30 years ago that at the present time can sound insulting. e.g. "women are less prepared in our educational systems for work life..." "Interesting, however, the gender gap in favor of males does not appear when looking at academic performance" Why is so interesting? "women have less resources, namely less time, energy and commitment..." Are you convinced of this? You are citing a reference from 1993. Life has changed a lot since them. Women have the same cited resources.
6) The educational system in Saudi Arabia is probably different to any other system form countries where the paper can contribute in the effort to bridge the gender gap. The system should be explained, or at least generally described.
7) What kind of institutions have been used for the research? Are they public institutions? Private institutions? Are those institutions gender mixed or they are male-female institutions? There is much information missing in this paper to be able to use the data in a comparative study.
8) To use actiotopes as influencers, the reader should know the content of those actiotopes, because the contexts of students in Saudi Arabia can be very different from other places in the world and therefore the data included in this paper may not be of help for future comparison studies.
Author Response
Dear Reviewer 2:
Thank you for your positive feedback and suggestions for improving the manuscript. Below is a list of your comments and the changes we have made as a result of your comments. Your comments have helped us a lot to optimize the manuscript.
1) From the title, it is not possible to know if the paper deals with higher education, secondary education, primary education... Should be clarified.
Thank you for this comment, we have changed the title and mentioned that it is a sample of secondary school students. We also included the term “secondary school” in the abstract and the keywords.
2) From the title, the reader would expect to find the analysis of the evolution (heuristic), but the experience is based on collection of data in 2019.
Thank you for this comment. To avoid misunderstandings, we deleted the turn of phrase that was misleading. We also made several other changes to our title to ensure that the reader knows that we will report an empirical study.
3) From the title, reversed gender achievement gap, one would expect to see a more positive achievement. From the text, it looks like there are optimistic data, but the gap is not explained as closed.
To make it explicit that we are examining the reversed gender achievement gap—and to avoid an over-optimistic assessment—we have changed the wording in the title and some relevant aspects in the manuscripts. We hope that this will help prevent misunderstandings.
After making the changes mentioned above the new title now reads: Correspondence Heuristic and Filter-Empowerment Heuristic: Investigating the Reversed Gender Achievement Gap in a Sample of Secondary School Students in Saudi Arabia within the Framework of Educational and Learning Capital
4) STEM includes very different areas. Literature shows that in general there is a higher success for male students in Maths and Physics, but in Biology the numbers are the contrary. If this is not the case in Saudi Arabia, a better explanation is needed.
Since we are studying a sample of secondary school students, we have used STEM as a conglomerate of the subjects science, technology, engineering, and mathematics, comparable to the PISA study (which covers a sample of a similar age group as ours). In our references on gender differences in STEM performance, we also refer to averaged performance of these subjects. Since the surveys in Saudi Arabia are comparable to those in other countries, this should not be a problem. To avoid misunderstandings, we have mentioned this at the appropriate place in the text.
5) The article includes assumptions from 30 years ago that at the present time can sound insulting. e.g. "women are less prepared in our educational systems for work life..." "Interesting, however, the gender gap in favor of males does not appear when looking at academic performance" Why is so interesting? "women have less resources, namely less time, energy and commitment..." Are you convinced of this? You are citing a reference from 1993. Life has changed a lot since them. Women have the same cited resources.
Thank you very much for these comments. We are sorry if any misunderstandings have arisen here. We are not stating our opinion, but quoting research. It is probably a linguistic problem. We have revised the relevant parts and added more recent literature. The following changes have been made:
"women are less prepared in our educational systems for work life" was changed to: Research suggests that women are less well prepared by their education systems for working life (Stotsky et al., 2016; World Economic Forum, 2022), for example, when it comes to salary negotiations (Hernandez-Arenaz & Iriberri, 2022; Säve-Söderbergh, 2019).
“women have less resources, namely less time, energy and commitment…”: This quote from 1993 does not reflect our conviction, but was meant to illustrate that lack of resources has already been used to explain why women on average are disadvantaged in science and at higher levels of seniority in professional life (not in secondary and higher education). Perhaps the misunderstanding was caused by the fact that we used a quote from 1993 that reads like an opinion. In our revised version of the manuscript, we have changed this and also cited more recent research. We hope to have cleared up the misunderstanding.
Specifically, we changed the following part of the manuscript: “For example, Toren (1993) explained gender inequality in academia by the fact that women have less resources, namely “less time, energy, and commitment to invest in their professional careers and are therefore less productive scientifically than men” (p. 72).” into “Similar to the correspondence heuristic, the filter-empowerment heuristic appears theoretically plausible and is supported by empirical research. For example, studies (e.g., Toren 1993, Ziegler et al., 2019) show that different resources are partly responsible for gender inequality in science and professional life. Specifically, it has been shown that women are on average less able to devote time, energy, and commitment to their careers (e.g., due to more extensive childcare and housework), which can lead to lower productivity.”
6) The educational system in Saudi Arabia is probably different to any other system form countries where the paper can contribute in the effort to bridge the gender gap. The system should be explained, or at least generally described.
The school system in the KSA is indeed different from many other school systems, especially in terms of gender segregation. We have included a paragraph in which we present the school system in the KSA in more detail. Specifically, we included the following paragraph:
In the KSA, children of both sexes attend nine years of compulsory education. Children between the ages of three and five can attend kindergarten. However, kindergarten attendance is not a prerequisite for admission to primary education. Primary education lasts six years; intermediate and secondary education last three years each (Unified National Platform GOV. SA., 2022). During their second year of secondary education, students choose a humanities or a natural sciences track. Schooling in the KSA is free, including textbooks (Faraj, 2005). The Ministry of Education issues and manages a unified curriculum and textbooks for all schools (Aljughaiman & Grigorenko, 2013). The total number of schools in the KSA is 22,986, of which 80.9% are public schools (ICEE, 2022). After completing secondary school, students can attend technical/vocational colleges or universities (Unified National Platform GOV. SA., 2022). Universities typically require students to pass benchmark tests such as a general aptitude test (similar to the SAT and ACT in the United States) and standardized academic achievement tests. Post-secondary education is also provided free of charge to Saudi citizens (Unified National Platform GOV. SA., 2022). As in the society as a whole, there is gender segregation in the higher education system. Educational institutions are, with few exceptions, either for men or for women (Aljughaiman & Grigorenko, 2013).
Although the school system in the KSA is different from many other school systems (but also similar to school systems in other Arabic countries), we believe that our study nevertheless, or precisely because of this, makes an important contribution to explaining the emergence of reversed gender gaps. One reason for this is that the KSA has a very strong reversed gender gap, which makes it particularly well suited to test the filter-empowerment heuristic and, more specifically, to examine the role played by the availability of learning and educational capital. To clarify this, we have made the following changes in the revised manuscript:
As mentioned earlier, although the correspondence heuristic has proved quite fruitful in many areas, it is limited in explaining the phenomenon of the reversed gender gap. In our view, the filter-empowerment heuristic appears more promising. The accuracy of this assumption can only be tested in a setting where there is a clear reversed gender gap. The situation in Arab countries is relevant in this context, especially when considering the Human Development Index (HDI).
Before describing the school system in the KSA, we have also included the following introduction:
We chose the KSA as the setting for our study. Its prevailing patriarchal social structures favoring men (Alexander & Welzel, 2011; Andersen et al., 2013) and the reversed gender achievement gap that has been demonstrated for the KSA in many studies (Bossavie & Kanninen, 2018) make it particularly suitable for examining the filter-empowerment heuristic (in contrast to the correspondence heuristic). Before going into more detail about our study, we will briefly describe the education system in the KSA.
Independently of this, we have pointed out generalization problems in the limitations and have made suggestions for future studies:
Furthermore, it is not clear to what extent the results of our study and the application of the filter-empowerment heuristic to the reversed gender achievement gap are transferable to other equity gaps, educational settings, and cultures. Future studies should try to replicate our findings in relation to other equity gaps and in other educational settings and cultures.
7) What kind of institutions have been used for the research? Are they public institutions? Private institutions? Are those institutions gender mixed or they are male-female institutions? There is much information missing in this paper to be able to use the data in a comparative study.
We included the following information about students and schools:
In order to test our hypotheses, we asked secondary students in the KSA to fill out a questionnaire. Participants were 2,541 ninth-grade students from 55 schools in Al-Ahsa, the largest governorate in the KSA’s Eastern Province. Students were on average 14.18 years old (range: 13–15 years, SD = 0.29); 61% were female; and 68% were from schools in urban areas. Twelve percent of students reported that their parents’ highest educational attainment was less than high school, 27% reported high school, 42% reported diploma or bachelor’s degree, 9% reported master’s degree, 4% reported a doctoral-level degree, and 5% of students did not provide valid information about their parents’ highest educational attainment.
Out of the 112 schools in Al-Ahsa, 55 schools were selected. To do so, a mix of stratified random sampling and convenience non-random sampling was utilized. The first step in sampling was to divide the schools into two subgroups based on the students’ gender (in the KSA schools are separated by gender). Next, these two subgroups were divided further into four subgroups, one based on the school location (urban or rural) and the other based on the type of school (private or public). Then a random sample was taken from each stratum (boys’ private school, boys’ public school, boys’ urban school, boys’ rural school, girls’ private school, girls’ public school, girls’ urban school, and girls’ rural school).
8) To use actiotopes as influencers, the reader should know the content of those actiotopes, because the contexts of students in Saudi Arabia can be very different from other places in the world and therefore the data included in this paper may not be of help for future comparison studies.
This is probably a misunderstanding. We do not consider actiotopes as influencers. Instead, we consider the learning and educational capital that is available to girls and boys and address the question of whether the availability of learning and educational capital contributes to explaining the reversed gender achievement gap within a culture, as assumed by the filter-empowerment heuristic. As stated above, we nevertheless point out in our limitations section that generalizability to other equity gaps, educational settings, and cultures should be examined in future research. The fact that we are concerned with differences in the availability of learning and educational capital within a culture and their explanatory contribution to the reversed gender achievement gap within this culture is presented in the following passage in the revised manuscript:
The filter-empowerment heuristic opens promising perspectives for tackling the problem of similarities and differences between individual equity gaps (Ziegler et al., 2021). This problem can now be concretized into a comparison of those resources available and used in the actiotopes of individuals that enable successful learning processes. To establish comparability, the education and learning capital approach of Ziegler and Stoeger (Ziegler & Baker, 2013; Ziegler & Stoeger, 2013; Ziegler et al., 2017) was used. Within the framework of the actiotope model, this approach can distinguish between the exogenous resources that enter a learners’ personal systems and the endogenous resources that are necessary for successful learning at school.
Although our study is primarily concerned with the influence of learning and educational capital on the reversed gender gap within one culture (the KSA) and although we have pointed out in our limitations section the problem of generalizability to other equity gaps, educational settings, and cultures, we assume on the basis of comparative cultural studies using the learning and educational capital approach that the influence of learning and educational capital on the reversed gender gap could be comparable in other cultures. It is precisely this fact that led us to choose the framework of learning and educational capital to test the filter-empowerment heuristic. This fact is presented in the revised manuscript in the following text passage:
The choice of the learning and educational capital approach was motivated by five considerations: (a) it specifies both endogenous and exogenous resources; (b) it is comprehensive and exhaustive (Vialle, 2017); it has already been successfully applied (c) in the target context of our study, the KSA, as well as in other cultures (Vladut et al., 2013), and it has been used (d) to explain gender differences; finally, (e) a measurement instrument based on the learning and educational capital approach validated in an Arab country was already available.
Round 2
Reviewer 2 Report
Authors have followed all the recommendations and have implemented the requested corrections to the text and title.